# Revealing polymerisation defects and formation mechanisms in aldol condensation for conjugated polymers via high-resolution molecular imaging

Xiaocui Wu[1,2], Stefania Moro[3], Adam Marks[4,5], Maryam Alsufyani[4,6], Zidi Yu[7], Luís M. A. Perdigão[1], Xingxing Chen[8], Alexander M. T. Luci[1], Callum Crockford[1], Simon E. F. Spencer [9], David J. Fox[1], Jian Pei [7], Iain McCulloch [4,10] & Giovanni Costantini [3,11] ✉

Aldol condensation is a crucial synthetic reaction in organic chemistry, particularly valued for fabricating conjugated polymers without the use of metals or toxic organostannanes. However, due to the lack of reliable and precise analytical methods, no direct evidence of the microstructure and sequence of synthesised polymers has been obtained, limiting control over their structure and performance. Here, by combining electrospray deposition and scanning tunnelling microscopy (ESD-STM), we analyse sub-monomer resolution images of four different n-type polymers produced via aldol condensation, revealing unexpected defects in both the sequence of (co)monomers and their coupling. These defects, observed across all polymer samples, indicate alternative side reaction pathways inherent to aldol condensation. Our findings not only uncover the reaction mechanism responsible for these defects but also bring new insights for the design of more effective synthetic pathways to minimise structural defects in conjugated polymers.

Conjugated polymers (CPs) present numerous advantages such as low cost, lightweight, mechanical flexibility and solution processability, making them particularly appealing materials for applications in optoelectronics, (bio)sensors, organic light emitting diodes (OLEDs), organic field-effect transistors (OFETs), neuromorphic computing and energy storage[1,2]. Since the report of poly(p-phenylene vinylene) for large-area OLED devices[3], a variety of CPs have been proposed and fabricated over the past 30 years with complex chemical compositions

and structures typically obtained through multi-step co-polymerisation techniques[4,5]. Cross-coupling polycondensation reactions catalysed by transition metals are commonly employed to synthesise CPs, including Suzuki–Miyaura[6,7], Mizoroki–Heck[8,9], Sonogashira[10], Kumada[11] and Stille polymerisation[12]. However, they often require the use of precious metals as catalysts, making them impractical for large-scale industrial use. As a consequence, more economical and environmentally friendlier alternatives have been sought after. Among

[1]Department of Chemistry, University of Warwick, Coventry, United Kingdom. [2]Institut Jean Lamour, CNRS, Université de Lorraine, Nancy, France. [3]School of Chemistry, University of Birmingham, Birmingham, United Kingdom. [4]Department of Chemistry, University of Oxford, Oxford, United Kingdom. [5]Department of Materials Science and Engineering, Stanford University, Stanford, California, USA. [6]Department of Chemistry, Massachusetts Institute of Technology, Cambridge, Massachusetts, USA. [7]College of Chemistry and Molecular Engineering, Peking University, Beijing, China. [8]Department of Materials, School of Chemistry & Chemical Engineering, Anhui University, Hefei, Anhui, China. [9]Department of Statistics, University of Warwick, Coventry, United Kingdom. [10]Andlinger Center for Energy and the Environment and Department of Electrical and Computer Engineering, Princeton University, Princeton, NJ, USA. [11]School of Physics and Astronomy, University of Birmingham, Birmingham, United Kingdom. ✉e-mail: g.costantini@bham.ac.uk

these, aldol condensation, one of the most versatile synthetic methods employed in organic chemistry[13], has attracted significant attention[14,15].

Aldol condensation involves the nucleophilic addition of a ketone enolate to an aldehyde or ketone. By losing a water molecule, which is the only by-product of the reaction, the initially formed β-hydroxycarbonyl compound is transformed into an α,β-unsaturated carbonyl species, which can then be extracted from the solution without laborious purification steps[16,17]. In contrast to the well-developed cross-coupling polycondensations, the aldol reaction does not require metal catalysts, which have been shown to often remain as residual impurities in the conjugated polymers and to affect their efficiency[18–20]. Moreover, aldol condensation affords rigid backbones due to the formation of double C=C bonds, thereby eliminating single-bond rotation and imparting a coplanar conjugated backbone, which is known to promote charge delocalisation and efficient charge transport[15]. Thus, it is emerging as a greener and cheaper approach to synthesise semiconducting polymers, in particular *n*-type materials[15,20–24]. However, limited information is available on the quality of the resulting materials, i.e., the precise chemical structure of the synthesised polymers in terms of their sequence and the potential presence of polymerisation defects, which are known to possibly affect the performance of CP-based devices[25–29].

There is therefore a rising demand for the precise and reliable characterisation of these functional materials, in particular of their exact sequence, which also allows for an improved understanding of the polymerisation mechanism and, ultimately, for a better, molecular-scale control of the synthesised materials. Traditional characterisation techniques such as nuclear magnetic resonance (NMR), mass spectrometry (MS) or size exclusion chromatography (SEC) are not ideal to analyse CPs given their low solubility, high tendency to aggregation, low volatility in MS and the lack of reliable calibration standards in SEC. Recently, the combination of electrospray deposition (ESD) with scanning tunnelling microscopy (STM) made it possible to determine

the exact sequence and mass distribution of the polymers, including the presence of possible polymerisation defects[25,30–33] as well as to establish the assembly patterns with molecular scale insight[34–37].

This methodology has been particularly useful in analysing the structure of the fused rigid-rod polymer **1** (Fig. 1), which is synthesised via aldol condensation and shows excellent environmental stability in OFET devices[38]. Besides demonstrating that this polymer has an unusually high persistence length, the ESD-STM characterisation also revealed the presence of kinks in the overall straight backbones, which were tentatively attributed to *cis*-defects in the double bond linkage between the monomers[35]. A simple statistical analysis of these kinks reveals that they are rather frequent, corresponding to about 9% of all monomer linkages (see section 4.2 of the Supplementary Information, SI). Because of the high rotational rigidity of double C=C bonds at room temperature, these kinks cannot result from conformational disorder but must be intrinsic structural defects arising from the way monomers have coupled during the polymerisation. Our preliminary investigations of **1** thus prompt a thorough reassessment of the conventional reaction mechanism typically assumed for aldol condensation.

To this aim, here we study four structurally different *n*-type copolymers of increasing complexity (Fig. 1) – all involving aldol condensation reactions. Specifically, while aldol condensation is used as the polymerisation reaction for copolymers **2**, **3** and **4**, in the case of copolymer **5**, aldol condensation is only employed to synthesise one of the two comonomers, which are subsequently polymerised through a Stille reaction. The ESD-STM analysis clearly identifies and quantifies the appearance of two types of unexpected polymerisation defects: deviations from the ideal polymer sequence, resulting from the wrong ordering of comonomers (sequence defects), and kinks in the backbone structure, resulting from their incorrect coupling position or orientation (coupling defects). Based on the experimental results, detailed reaction pathways leading to the formation of the observed

**Fig. 1 | Chemical structures of the conjugated polymers studied in this work.** The comonomers of each polymer are marked in a different colour and labelled with a different letter. The generic aldol condensation steps involved in the synthesis of polymers **1**–**5** are shown in Supplementary Fig. S1, and the polymerisation steps used to synthesise polymer **5** are shown in Supplementary Fig. S2.

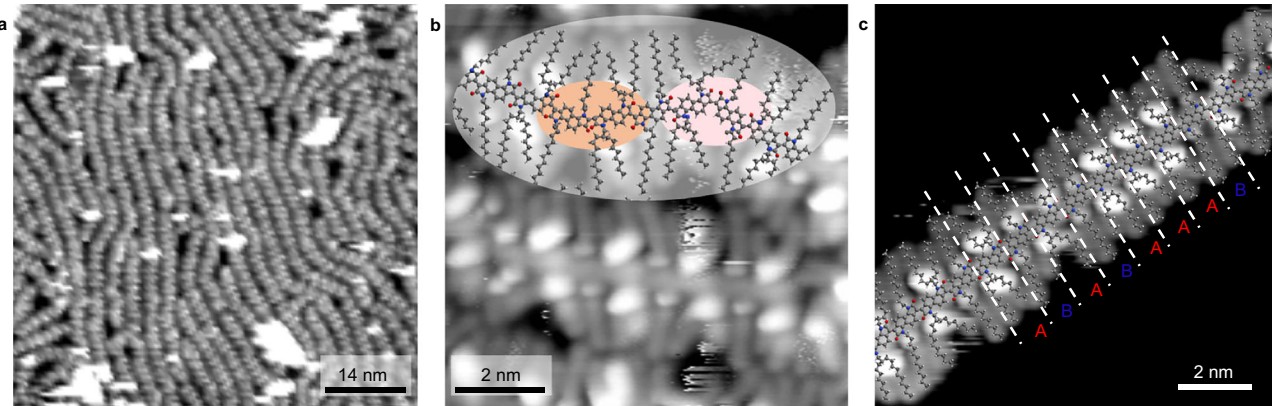

**Fig. 2 | STM characterisation of polymer 2. a** Large-scale STM image of copolymer **2** acquired in a high coverage area (70 nm × 70 nm, V = 950 mV, I = 78 pA). **b** High-resolution STM image of a high coverage area (10 nm × 10 nm, V = 51 mV, I = 90 pA) showing coupling defects. Pink and orange ovals highlight different types of kinks in the polymer backbone and correspond to c2 and c3 defects, respectively, as illustrated in Fig. 3a. **c** High-resolution STM image of an isolated polymer showing sequence defects (10 nm × 10 nm, V = 149 mV, I = 120 pA). Letters A and B refer to the two comonomers of polymer **2** defined in Fig. 1. Scaled molecular models of the polymer are superimposed on the upper part of image (**b**) and on image (**c**).

defects are proposed, bringing new insight into the mechanism of the aldol condensation reaction.

## Results and discussion

The first step in our study was to move from homopolymer **1** by increasing its complexity while also exploring the potential role of bulky side chains in influencing the occurrence of coupling defects. To this aim, we synthesised co-polymer **2** (see Fig. 1), which has a branched $C_2C_{8;10}$ (with $C_2$ being the linker to the branched $C_{8;10}$ side chain) and a linear $C_{12}$ alkyl side chain tethered to the bis-oxindole monomer units. Effectively, **2** is obtained from **1** by replacing every second $C_2C_{8;10}$ branched side chain with a less bulky linear $C_{12}$ chain.

Figure 2 shows STM images acquired after the room temperature ESD of **2** onto a pristine Au(111) surface. In high molecular coverage areas, the polymers form relatively compact monolayers where individual molecules tend to align mostly parallel to each other (Fig. 2a). Individual polymers (Fig. 2c and Supplementary Fig. S5a) have straight backbones (grey stripe in Supplementary Fig. S5b) surrounded on both sides by alternating larger and smaller oval-shaped bright dots (red and blue in Supplementary Fig. S5b, respectively), and by dimmer elongated features emerging almost perpendicularly from these dots. These are identified as the side chains, with the larger dots corresponding to the starting points of the branched $C_2C_{8;10}$ alkyl side chains, and the smaller ones representing the starting points of the linear $C_{12}$ side chains.

While the backbones of **2** are mostly straight, occasional kinks can be seen in the STM images (e.g., see pink and orange ovals in Fig. 2b and Supplementary Fig. S6), similarly to what had been observed for **1**[35]. In order to determine the precise molecular structure of the polymers in correspondence to these kinks, we fitted geometry-optimised molecular models of **2** onto a large number of STM images (e.g., Fig. 2b), making sure that all features in the images were correctly accounted for, in particular the positions, orientations and starting points of the side chains (see section 4.1 of the SI for a detailed explanation of the fitting process). By doing so, we were able to identify several types of coupling between the A and B comonomers,

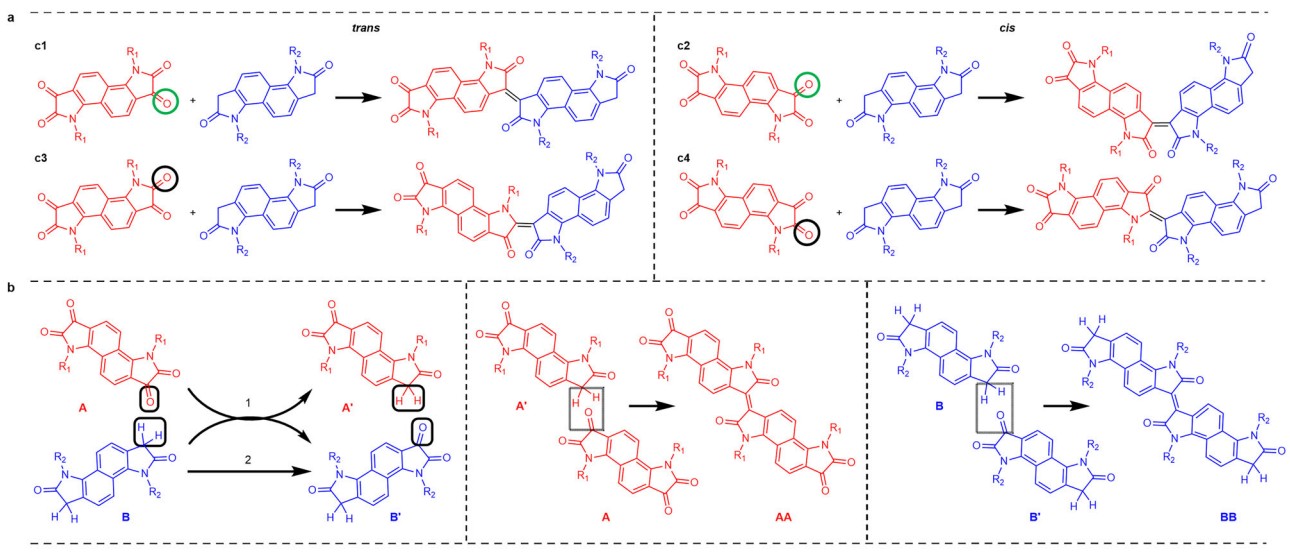

1. Interconversion via acid catalyse hydride transfer; 2. Aerial oxidation.

**Fig. 3 | Possible reaction pathways for coupling and sequence defects between two isatin-based comonomers. a** Coupling defect. Reactions c2-c4 lead to the formation of kinks in the polymer backbone, with progressively shallower angles. The carbonyls β and α to the nitrogen atom of the pyrrole ring, where the reaction takes place, are circled in green and black, respectively. **b** Possible reaction pathways leading to AA and BB polymer sequence defects. A and B comonomers are marked in red and blue, respectively.

**Table 1 | Backbone defect frequencies for polymers 2 (left table) and 3 (right table)**

| Coupling | Count | Frequency | Coupling | Count | Frequency |
|---|---|---|---|---|---|
| c2 | 19 | 2.6% | c2 | 20 | 3.6% |
| c3 | 12 | 1.7% | c3 | 4 | 0.7% |
| c4 | 0 | 0.0% | c4 | 2 | 0.4% |
| Sum | 31 | 4.3% | Sum | 26 | 4.7% |
| c1 | 687 | 95.7% | c1 | 525 | 95.3% |
| **Sequence** | **Count** | **Frequency** | **Sequence** | **Count** | **Frequency** |
| AA | 10 | 1.4% | AA | 16 | 2.9% |
| BB | 15 | 2.1% | BB | 66 | 12.0% |
| Sum | 25 | 3.5% | Sum | 82 | 14.9% |
| AB | 693 | 96.5% | AB | 469 | 85.1% |
| **Monomer** | **Count** | **Frequency** | **Monomer** | **Count** | **Frequency** |
| A | 436 | 51.6% | A | 327 | 47.1% |
| B | 409 | 48.4% | B | 367 | 52.9% |

While c1 is the correct *trans* coupling, c2-c4 are defective couplings, as illustrated in Fig. 3a. The relative frequencies were calculated with respect to the total amount of analysed coupling and sequence configurations. The corresponding 95% confidence intervals are reported in Supplementary Table S1.

differing from those typically expected in aldol condensation reactions, offering a potential explanation for the origin of the observed kinks (Fig. 3a).

The most frequently observed coupling (as reported in Table 1) is c1 in Fig. 3a, where the carbonyl β to the nitrogen in the A comonomer (circled in green in Fig. 3a) reacts with the oxindole hydrogen of the B comonomer. This *trans* coupling is what is usually expected in the aldol condensation reaction and results in a straight polymer backbone. However, the same two comonomers could potentially also react in the c2 *cis* coupling geometry in Fig. 3a. This latter reaction is expected to be less favourable than c1 because of the steric repulsion from the close proximity of the two carbonyls in the final product, which causes a kink of about 130° in the polymer backbone (Supplementary Fig. S7). The c2 *cis* backbone configurations are indeed observed in the STM images (e.g., see pink oval in Fig. 2b and Supplementary Fig. S6a), although much less frequently than the straight *trans* configurations (see Table 1). Very similar kinks were also observed for 1[35] (see also Supplementary Fig. S4), so we expect them to be caused by the same type of c2 *cis* coupling.

The bis-isatin A comonomer of **2** has two carbonyl groups. The reactivity of the carbonyl β to the nitrogen atom (circled in green in Fig. 3a) is expected to be higher due to a combination of steric and electron density arguments. In fact, the β carbonyl is more susceptible to attack from the enolate, which is formed upon deprotonation of the acidic α hydrogen on the oxindole monomer. However, it cannot a priori be excluded that also the carbonyl α to the nitrogen atom (circled in black in Fig. 3a) may react with the oxindole hydrogen of the B comonomer. Moreover, for this reaction, both *trans* and a *cis* configurations are in principle possible, resulting in the two further coupling schemes c3 and c4 in Fig. 3a, respectively. Based on simple molecular models, the backbone configurations resulting from c3 and c4 are expected to show less sharp kinks of about 150° and 175°, respectively (Supplementary Fig. S7). In the STM images, only very few cases of c3 couplings could be identified (e.g., see orange ovals in Fig. 2b and Supplementary Fig. S6b), while no c4 coupling could be observed.

Model reactions for polymer **2** have been reported without significant side products[38]; while this may reflect analytical limitations in detecting low-abundance byproducts, the much lower solubility of the polymer compared to its monomeric precursors means that reaction equilibria in model systems are unlikely to reflect those of the actual polymerisation. In fact, the equilibria of most chain polymerisations and polyeliminations are not governed by complete equilibrium among all components[39]. As a result, model reactions are not expected to reliably represent the frequency of coupling defects in the polymer. Only direct structural analysis of the polymers themselves, as performed here, can provide this level of insight.

While polymer **2** is mostly characterised by an alternating (AB)ₙ sequence where the side chains in consecutive comonomers are different, the analysis of the STM images also identified several cases where two (or even more) identical side chains are located in successive positions on both sides of the backbone. An example can be found in Fig. 2c, showing an isolated polymer strand where the otherwise alternating sequence of A and B comonomers is interrupted by three successive A units in its upper right section. Another example is given in Supplementary Fig. S8, displaying a case of two successive B units.

These configurations cannot result from direct homocouplings between the A and B comonomers: in fact, the A comonomer cannot couple to itself by aldol condensation, while a *trans* BB homocoupling would create a noticeable kink in the polymer backbone (Supplementary Fig. S9); a *cis* BB homocoupling would cause a much shallower kink but would also have two successive side chains on the same side

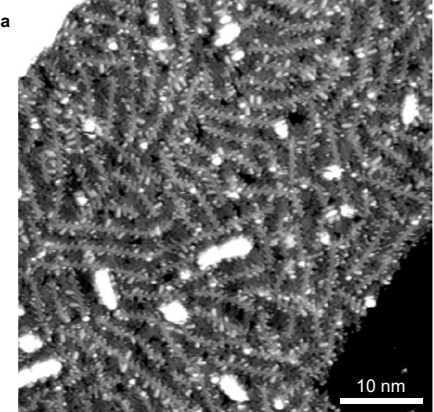

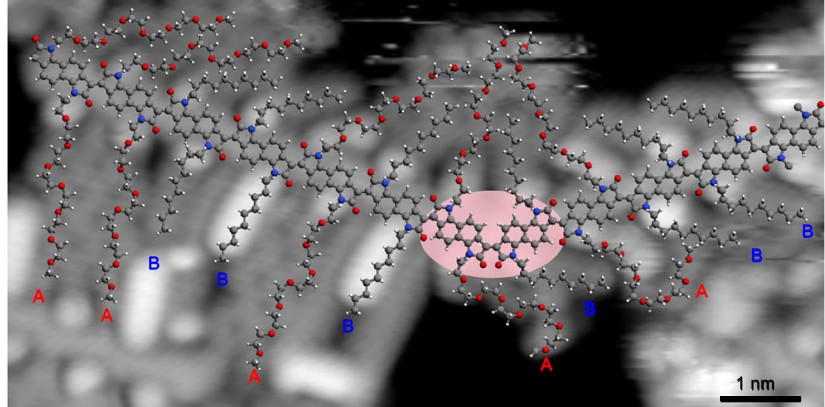

**Fig. 4 | STM characterisation of polymer 3. a** Large-scale STM image (50 nm × 50 nm, $V = 950$ mV, $I = 110$ pA) of copolymer **3**. **b** High-resolution STM image showing defects in the polymer sequence and backbone structure (5 nm × 10 nm, $V = 247$ mV, $I = 140$ pA). The kink in the backbone is marked by a pink oval and corresponds to a c2 (*cis*) coupling (less frequent cases of c3 and c4 couplings are shown in Supplementary Fig. S10). A geometry-optimised molecular model of the polymer is overlayed on the image in (**b**). Letters A and B refer to the two comonomers of polymer **3** defined in Fig. 1.

Polymer **4**        Polymer **5**

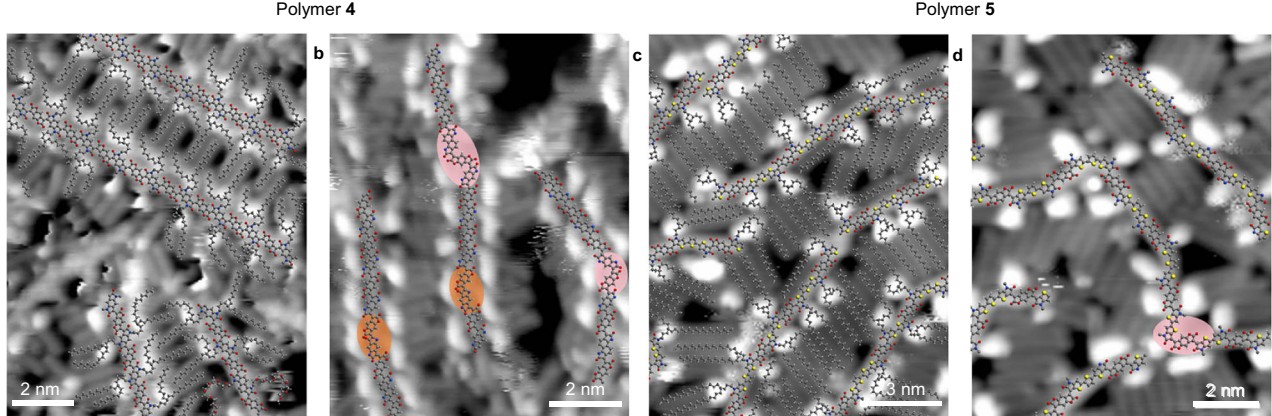

**Fig. 5 | STM characterisation of polymer 4 and 5. a** Sequence of polymer **4** with predominant AB coupling (15 nm × 15 nm, $V = 573$ mV, $I = 78$ pA). **b** Formation of kinks in the backbone (10 nm × 10 nm, $V = 176$ mV, $I = 90$ pA). Pink and orange ovals correspond to c2 and c3 coupling defects, respectively, as illustrated in Supplementary Figs. S12 and S13. **c** High-resolution STM images of polymer **5** (13 nm × 13 nm, $V = 247$ mV, $I = 140$ pA). **d** Formation of kinks in the backbone (10 nm × 10 nm, $V = 247$ mV, $I = 140$ pA). The pink oval corresponds to a c2 defect (see Supplementary Fig. S17). Geometry-optimised molecular models are superposed onto images in (**a**) and (**c**), while those of only the backbone are superposed onto images in (**b**) and (**d**) to help visualise the kinks.

of the backbone (Supplementary Fig. S9). Neither kinks nor same-side side chains were ever observed in the case of successive identical side chains. On the contrary, AA and BB polymer sequences were always seen to occur in straight segments of the backbone, indicating that they are resulting from a "regular" c1 coupling. Thus, these sequence defects are likely the consequence of a different reaction mechanism. One possibility is for the $CH_2$ in A and the $C=O$ in B to exchange positions in an acid-catalysed hydride transfer reaction[40], producing A' and B', thus allowing the formation of both AA and BB units in the polymer (Fig. 3b). The proposed mechanism of this interconversion is shown in section 3 of the SI. In addition, it is also possible to oxidise (via aerial oxidation during the aldol reaction) the $CH_2$ in monomer B to a $C=O$, forming B', allowing its combination with a non-oxidised B to give a BB unit (Fig. 3b). This additional potential pathway is likely the underlying reason for the higher frequency of BB compared to AA defects (Table 1).

In order to check generality of the different reaction pathways identified for **2** and to gauge the influence of the side chain chemistry on these reaction mechanisms, we analysed **3** (Fig. 1), obtained by replacing the branched hydrophobic alkyl side chain of **2** with a linear hydrophilic 7-unit ethylene glycol (EG) chain (henceforth identified as $g_7$), and the linear $C_{12}$ alkyl chain by a slightly shorter $C_{10}$ chain. This polymer was also made by aldol condensation and shows high electron mobility in organic electrochemical transistors (OECTs)[41].

Figure 4a shows that polymer **3** self-assembles into extended compact islands, which, despite being less ordered than those of **2**, still present local parallel packing. Close-up images (Fig. 4b) allow us to identify a characteristic sub-structure with bright alternating smaller ovals and larger dumbbell features on both sides of the backbones (highlighted in blue and red, respectively, in Supplementary Fig. S11b). Shorter linear structures (blue lines in Supplementary Fig. S11b) extend from the smaller dots and are associated with $C_{10}$ alkyl chains, while longer serpentine features (red lines in Supplementary Fig. S11b) emerge from the dumbbells and are assigned to the $g_7$ EG side chains.

Figure 4b demonstrates that, also for **3**, coupling and sequence defects can be identified in the STM images, resulting in kinked backbones and in occasional AA and BB sequences. As in the case of **2**, the sequence defects occur only in straight segments of the backbone, excluding the possibility of homocouplings, but instead pointing to the effective exchange of functional groups between comonomers as proposed in Fig. 3b. Moreover, fitting the STM images of **3** in correspondence to the kinks, reveals the presence of the "incorrect" couplings c2, c3 and c4 shown in Fig. 3a (Supplementary Fig. S10).

Since **2** and **3** exhibit similar polymerisation defects, it raises the question of whether their side chain chemistry affects the kinetics of the aldol condensation reaction, potentially altering the occurrence of defects. To address this, a statistical analysis of backbone defects was conducted for both polymers and is summarised in Table 1.

The results show that polymers **2** and **3** have a similar amount of coupling defects, with a total percentage of "incorrect" couplings (c2, c3 and c4 in Fig. 3a) of 4-5%, with c2 being the predominant defective coupling. Thus, these sets of data demonstrate experimentally that the different side chains of **2** and **3** do not significantly alter the relative reactivity of the two carbonyls in the A bis-isatin comonomers or the relative probability of *cis* or *trans* couplings. This outcome is somewhat anticipated because, although the oxygen atoms in the EG side chains of **3** might have the potential to affect the electronics/acidity of the carbonyl groups at the α and β positions in the bis-isatin monomer, they are situated at a considerable distance. In addition, the steric hindrance that impairs the defective couplings is mostly independent of the side chains in c2 and c3, and just dependent on the presence of a side chain in c4, rather than on its specific chemical nature.

However, an obvious difference is observed when it comes to the sequence defects. In fact, while for **2** only a limited amount of sequence defects is detected (with 1% of AA and 2% of BB sequences, see Table 1), these are much more frequent for **3** (with 3% of AA and 12% of BB sequences). It is important to notice that, for both polymers, the total relative amounts of the two comonomers observed in the STM images is very close to the expected value of 1:1, confirming that the correct relative stoichiometry was used for the polymer synthesis. The reported difference seems, therefore, to be intrinsic to the aldol condensation reaction and suggests a marked influence of the side chain chemistry on the kinetics of the reactions described in Fig. 3b.

To further our analysis of how different chemical and structural parameters influence the generation of synthetic defects during aldol condensation polymerisation, we decided to study **4**, a recently proposed lactone-based rigid semiconducting polymer[24], whose synthesis involves also a third, lactone-based comonomer without side chains (Fig. 1). Thus, **4** is characterised by a different chemistry of the backbone, by a lower side chain attachment density with respect to **2** and **3** and by longer and more flexible branched alkyl side chains when compared with **1** and **2**.

High-resolution STM images (Fig. 5a and Supplementary Fig. S14) show that the polymer backbones are predominantly functionalised with alkyl branched side chains – $C_{10;12}$ with linear $C_5$ spacers (red lines

in Supplementary Fig. S14d) – with only a few cases of EG $g_7$ side chains observed (magenta lines in Supplementary Fig. S14d). Thus, the ESD-STM analysis demonstrates that the structure of **4** is not the expected $(AB)_m(CB)_n$ sequence (Fig. 1) but, mostly, an alternating $(AB)_n$ sequence, which is probably due to the higher solubility of the A comonomer with respect to C in toluene (see section 5 of SI).

The occurrence of occasional kinks is also observed in the backbones of **4** (pink and orange ovals in Fig. 5b). These correspond to coupling defects between both AB and BC pairs of comonomers, resulting from two different sets of possible reaction pathways (Supplementary Figs. S12 and S13, respectively), which are the isatin-lactone equivalents of the isatin-isatin reactions shown in Fig. 3a. Examples of c2 and c3 couplings, observed both between AB and BC comonomers, are shown in Supplementary Fig. S16, while c4 couplings were only observed between AB comonomers. A quantitative analysis of the coupling defects reveals a total frequency of 3.7% (Supplementary Table S2), which is comparable to those obtained for **2** and **3** (Table 1). This confirms our previous finding that the relative reactivity of the two carbonyls in the bis-isatin-based comonomer does not depend significantly on the chemical nature or, in this case, on the absence of its side chains, nor on whether the isatin monomer is reacting with another isatin-based or with an oxindole-based comonomer. Finally, we note that **4** cannot develop any of the sequence defects as those observed for **2** and **3**, because for **4** the aldol condensation occurs between an isatin-based and an oxindole-based comonomer, and thus the effective comonomer exchange reactions described in Fig. 3b cannot take place. No such type of defect was in fact observed in the STM images.

As a final effort to assess the generality of our findings on defects in the aldol condensation, we analysed a last polymer, **5**[42–44]. Its synthesis involves both an aldol condensation reaction to produce the TBDOPV comonomer and a Stille polymerisation to bridge the TBDOPV unit with the thiophene unit (Fig. 1). Thus, **5** is characterised by a different chemistry of the backbone, by a lower side chain attachment density with respect to **1-3** and by even longer and more flexible branched alkyl side chains when compared with **1**, **2** and **4**.

No sequence defects of the type described for **2** and **3** were ever observed within the TBDOPV moieties of polymer **5** (Fig. 5c, d). As in the case of **4**, this is attributed to the presence of the lactone-based unit, which prevents the exchange reactions outlined in Fig. 3b. However, high-resolution STM images (Fig. 5d and Supplementary Fig. S18) reveal a small number of kinks (just above 1%) within the TBDOPV segments, likely resulting from "wrong" couplings. The corresponding proposed reaction pathways are illustrated in Supplementary Fig. S17, with the detailed defect statistics provided in Supplementary Table S3. NMR and single-crystal X-ray diffraction analyses of the TBDOPV monomers used to synthesise polymer **5** showed no evidence of coupling defects[45], consistent with their purification by silica gel chromatography and subsequent HPLC, both of which are expected to effectively separate *trans* and *cis* isomers in small molecules.

Whether the 1.3% total coupling defects observed in the STM images originate from trace impurities in the monomer (below the detection limit of NMR) or are introduced during the Stille polymerisation step (e.g., due to temperature or acidic/basic conditions) cannot be conclusively determined. However, the key insight is that the defect frequency is substantially lower when aldol condensation is used to prepare small-molecule comonomers (as in polymer **5**) than when it is employed directly for polymerisation (as in polymers **1–4**).

We attribute this difference to the effectiveness of chromatographic purification for small molecules, where *trans* and *cis* isomers differ significantly in polarity and can thus be efficiently separated. In contrast, defective polymer chains are not sufficiently distinct in polarity from their defect-free counterparts to enable separation by standard purification techniques. Thus, while defects likely occur

during both small-molecule and polymer-forming aldol condensation reactions, only in the former can they be efficiently removed. In the latter case, they remain embedded in the final polymeric product. These defects could not be previously detected, even when present at levels around 5%, as conventional analytical methods cannot resolve the precise sequence and composition of conjugated polymers. Our study, using ESD-STM, makes such analysis possible.

Finally, we note that an additional class of polymerisation defects was observed in polymer **5**, corresponding to homocoupling events occurring during the Stille reaction cycle both between TBDOPV and thiophene comonomers (see section 4.6 of the SI for more details). These defects are synthetically more closely related to the sequence defects we previously reported in conjugated polymers produced via cross-coupling reactions[25,30,33,46].

To summarise, in this work, we have employed the recently developed ESD-STM technique to precisely characterise the chemical structure of four *n*-type co-polymers of increasing complexity (**2–5**), all synthesised via aldol condensation. The high-resolution STM images provide unprecedented insights into the microstructure of these conjugated polymers, revealing the presence of so-far unaccounted defects in the backbones of all four polymers. These defects manifest as deviations from the expected polymer sequence (sequence defects) and as kinks in the backbone structure (coupling defects). A quantitative analysis unveils a non-negligible number of defects in all polymers synthesised by aldol condensation. These findings strongly imply that both types of defects are intrinsic to the aldol condensation reaction itself and highlight that the actual structure of the synthesised molecules can significantly deviate from the expected "ideal" structure. Moreover, it is found that, while coupling defects depend only weakly on the chemical nature and bulkiness of the side chains, the latter have a significant influence on the sequence defects.

Based on the experimental results, detailed reaction pathways elucidating the formation of the observed defects are proposed. These include, for the coupling defects, reactions occurring at alternative coupling sites of the comonomers both in *trans* and *cis* configurations, and, for the sequence defects, an acid-catalysed hydride transfer reaction causing a functional group interconversion between comonomers. Our analysis also shows that sequence defects do not occur in **4** and **5** since the incorporation of side-chain-free lactone-based comonomers effectively quenches functional group exchange reactions.

The results of this work bring new insight into the mechanism of the aldol condensation reaction and offer a rational basis for future design and improvement of new polymerisation methods to better control the polymer structure and thus their performance in optoelectronic applications. Moreover, this study demonstrates the prevalence and drastic impact of defects on the sequence and structure of conjugated polymers made by aldol condensation and highlights that care should be taken when assuming a perfect alternating repeating sequence of comonomers arranged in a straight, all-*trans* configuration. Future work should aim to optimise reaction conditions and monomer composition to deter and hopefully eliminate the unwanted defects. This would eventually fully realise the promise of the aldol condensation reaction as a much greener and sustainable alternative to current synthetic strategies for conjugated polymers, with no toxic byproducts and without the requirement of expensive rare metal catalysts.

## Methods
### Materials

The synthetic details for polymers **1-2** are described by Onwubiko et al.[38], and those for polymers **3–5** by Marks et al.[41], Alsufyani et al.[24] and Lu et al.[42], respectively. The polymers analysed in this work were synthesised according to these protocols which are described again in detail in the section 2 of the SI.

## STM characterisation

The STM experiments were performed in an ultra-high vacuum (UHV) low-temperature (LT) STM system (CreaTec Fischer & Co. GmbH). Atomically clean and flat Au(111) surfaces were prepared by cycles of Ar$^+$ ion sputtering ($p_{Ar} = 1.0 \times 10^{-5}$ mbar, 10 min) and annealing (400 °C, 10 min) and were checked by STM before being exposed to the polymers. All polymers were initially dissolved in chlorobenzene at the same concentration (0.05 mg/ml) and sonicated for 20 min (or longer if necessary) to ensure the molecules dissolved completely. Then, 1.7 ml of dissolved polymer solution was mixed with 0.3 ml of methanol, and the mixture was deposited by ESD (Molecularspray Ltd.) onto the pristine Au(111) surface held at room temperature. After deposition, the sample was cooled down by a liquid nitrogen flow cryostat and transferred to the STM analysis chamber for in situ imaging under UHV at low temperature (−196 °C in a bath cryostat). All the STM images were acquired in constant current mode and analysed by WSxM[47]. LMAPper[48] was used to fit the STM images with scaled molecular models of the polymers, which had previously been geometry optimised by the MMFF94 force field in the Avogadro software.

## Data availability

Data sets generated during the current study are available from the corresponding author on request.

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

## Acknowledgements
The authors thank Stephan Rauschenbach, Sarah Horswell and Stephen Goldup for insightful discussions. X.W. acknowledges co-funding from the European Union's Horizon 2020 research and innovation Marie Skłodowska-Curie Actions, under grant agreement no. 945380. G.C. and S.M. acknowledge support from a UK–Saudi Challenge Fund grant from the British Council's Going Global Partnerships programme. M.A. and I.M. acknowledge financial support from KAUST Office of Sponsored Research CRG10, by EU Horizon2020 grant agreement n°952911, BOOSTER, grant agreement n°862474, RoLA-FLEX, and grant agreement n°101007084 CITYSOLAR. X.C. acknowledges support from the National Natural Science Foundation of China (Grant NO.22205004), as well as EPSRC Projects EP/T026219/1, EP/W017091/1, and EP/L011972/1.

## Author contributions
X.W. performed STM measurements on polymers **2**–**5**, analysed the data and wrote the first draft of the manuscript. S.M. provided guidance on STM measurements and data analysis and contributed to data analysis and interpretation. C.C. contributed to the analysis of STM data for polymer **3**. L.M.A.P. carried out STM measurements on polymer **1** and analysed the data. A.M.T.L. assisted with STM measurements on polymer **1**. S.E.F.S. performed the statistical analysis of polymer defect frequencies and their associated uncertainties. A.M. synthesised polymer **2** and A.M. and X.C. synthesised polymer **3**. M.A. synthesised polymer **4** and conducted variable temperature ¹H NMR experiments. Z.Y. synthesised polymer **5**. D.J.F. proposed the $C=O$ / $CH_2$ interconversion mechanism. J.P. supervised the synthetic work for polymer **5**, while I.McC. supervised the synthesis of polymers **1**–**4**. G.C. conceived and supervised the overall project. All authors contributed to the editing and revision of the manuscript.

## Competing interests
The authors declare no competing interests.
