## [Transparent Peer Review file · Nature Communications]

Revealing polymerisation defects and formation mechanisms in aldol condensation for conjugated polymers via high-resolution molecular imaging

Corresponding Author: Professor Giovanni Costantini

Version 0:

Reviewer comments:

Reviewer #1

(Remarks to the Author)

In this work, the authors present the ESD-STM investigation of n-type conjugated polymers synthesized by aldol polycondensation from rigid oxindole monomers. They aim to elucidate the structural defects induced during the aldol reaction, resulting in the typical sequence defects and coupling defects. The underlying reaction mechanism for defect formation during the aldol reaction is proposed. Overall, the understanding of the polymerisation defects and formation mechanism for the studied conjugated polymers is interesting and could provide important insights into such particular classes of chemistry and materials. It is a pity that the defect structure-property relationship (e.g. FET mobility) is not provided for the studied polymers, which would strengthen the impact of the current work. In any case, I feel that the chemistry part needs to be further strengthened and the following points need to be carefully considered:

-Since the C2 defect is the rotational conformation isomer of the C1 unit through the C=C bond, it could also be formed during deposition from solution on the surface. What is the rotational energy barrier between C1 and C2?

-The difference in reactivity between alpha- and beta-carbonyl (as demonstrated in C3 and C1) can be expected for the aldol reaction. Have the authors studied the model reactions? I think this would explain the reactivity and selectivity for the particular oxindole monomers, which also helps to explain the coupling defect ratio in the resulting polymers.

-The observation of C=O/CH₂ interconversion via hydride transfer is indeed interesting for the current work. I would suggest that the authors look at the model reactions, which will be very helpful in understanding how the chemistry works. And what about the efficiency of this interconversion in relation to the reaction conditions (acidic or basic condition)?

-The details of the polymer synthesis (4 and 5) should be given in the main text or in the SI. The ¹H NMR spectra for polymer 4 appear to be of poor quality. There are too many dirty peaks and the assignment of the peaks is not clear.

Reviewer #2

(Remarks to the Author)

In this paper, the authors demonstrate the strength of using STM to characterize semiconducting polymers to give quantitative analysis of the synthesized products. Given that techniques such as NMR can only give an analysis of the average, STM offers a significant advantage. For this reason, I believe that this paper is appropriate for publication in Nature Communications.

However, I do have one major concern. The aldol reaction, unlike the cross-coupling reactions that the authors have investigated before, is a reversible reaction. Given the use of high fields during electrospray deposition, how confident are the authors that the results seen by STM are from the synthesis and not of additional reactions that took place during deposition? Without confirming this, any analysis of the polymers would seem premature.

Here are additional minor comments:

1. A super minor point – on page 8, it is noted that 3 was obtained by replacing the branched hydrophobic alkyl side chain of 2 with a linear hydrophilic 7-unit ethylene glycol (EG) chain. One other change is that the linear C12 chain has been replaced by a C10 chain according to Figure 1.

2. For the experimental details of the synthesis of the polymers, the authors refer to previously published work. However, it would be helpful to have the details in this paper too to confirm aspects of their findings. Just as one example, on page 11,

the characterization of polymer 4 indicates that comonomer C was not incorporated very much. It would be helpful to immediately have in hand what monomer ratios were used in the synthesis.

3. Page 11: Can there be a table quantifying the defects in polymers 4 and 5, in the same way that there is one for polymers 2 and 3?

4. Page 11: The NMR shows that there is ~36% of the hydrophilic monomer. I know that there is error in the NMR analysis but how does this compare against the STM results?

5. Section 4 in SI: Based on the trends in solubility, I would have expected the polymer synthesized in aniline to have the greatest hydrophilic monomer content. Can the authors elaborate on this a bit more? The paper would also be greatly strengthened if the STM results for the polymers synthesized in aniline and aniline:toluene could be included.

6. Page 12: "This finding demonstrates that this type of coupling defects is not limited to polymerisation reactions, as is seen in the case of polymers 1-4, but can also occur during the synthesis of small molecules, as seen with polymer 5." For the defects occurring during monomer synthesis, it should be possible to see these in the ¹H NMR (if it can't be seen in 1D NMR, it should be possible to see it in the 2D NMR). Can the statement be confirmed with further analysis? Given that the aldol reaction is reversible, it would be good to check that the defect formed during monomer synthesis instead of during the polymerization when decoupling followed by coupling could have taken place.

Version 1:

Reviewer comments:

Reviewer #1

(Remarks to the Author)

The authors have mostly addressed my concerns in the revised manuscript. However, it is a pity that they did not follow the suggestion for model compound studies. I understand that this would require considerable effort. In light of this, I wonder if the authors could provide computational studies of the reactions or intermediates, which could help to clarify the reactivity and selectivity related to C2 and C3 coupling defects.

Reviewer #2

(Remarks to the Author)

The authors have addressed the concerns raised. I believe that the paper is acceptable for publication .

Response to reviewers' comments

We thank the Reviewers for their thorough evaluation of our manuscript. We are very grateful for their insightful feedback, which has allowed us to further clarify and strengthen key aspects of our work, resulting in what we believe is now a significantly improved version of the paper.

We appreciate that both Reviewers recognise the value of our ESD-STM analysis of *n*-type conjugated polymers synthesised via aldol polycondensation. Reviewer #1 highlights that our work “could provide important insights into such particular classes of chemistry and materials,” while Reviewer #2 acknowledges that our ESD-STM analysis “offers a significant advantage”, believing “that this paper is appropriate for publication in Nature Communications.”

While both Reviewers express a positive overall assessment, they have offered a number of comments and suggestions. We address each of these points in detail below. The corresponding modifications to the main text and the Supplementary Information have been highlighted in blue font.

Reviewer #1

Since the C2 defect is the rotational conformation isomer of the C1 unit through the C=C bond, it could also be formed during deposition from solution on the surface. What is the rotational energy barrier between C1 and C2?

The rotation around the C=C double bond is energetically extremely demanding and we believe it cannot be induced during the deposition process. Unlike C-C single bonds, C=C double bonds are rigid due to the presence of the π bond and “rotations” about them are associated with a high energy barrier. While the precise rotational barrier depends on the chemical environment, a useful benchmark is provided by the *cis-trans* isomerisation of but-2-ene, which has an experimentally determined energy barrier of approximately 65 kcal/mol [M. C. Lin and K. J. Laidler, *Can. J. Chem.* 46, 973 (1968) and references therein]. This value needs to be compared with the kinetic energy of the polymers upon deposition on the Au(111) surface.

Estimating the impact kinetic energy of molecules during electrospray deposition (ESD) is non-trivial, particularly because the vast majority of molecules are expected to land on the surface still solvated within microdroplets. These droplets are produced through the electrospray process, originating from the disruption of the Taylor cone and undergoing successive Coulomb explosions. Driven by the electric field established by the high voltage applied between the emitter needle and the entrance capillary, both the droplets and a co-flowing stream of ambient air are accelerated toward the capillary. This capillary serves as the interface between atmosphere and the first vacuum stage of the differential pumping system of the ESD source. Upon entering the capillary, the droplet-air mixture begins to expand into vacuum. In particular, the air molecules are expected to undergo an isentropic expansion, accelerating towards a terminal velocity given by:

$$v_{\infty} = \sqrt{\frac{2\gamma k_B T}{(\gamma - 1)m}}$$

where γ is the adiabatic index, k_B is the Boltzmann constant, T is the nozzle temperature ($T \sim 300$ K in our experiments) and m is the molecular mass. Assuming for simplicity that air consists solely of N_2 molecules ($\gamma_{N_2} \sim 1.4$ and $m_{N_2} = 4.68 \times 10^{-26}$ Kg), we obtain $v_{\infty} = 787 \text{ ms}^{-1}$ (corresponding to Mach 2.3).

In typical molecular beam experiments, the species of interest are present under extremely low concentrations in the carrier gas. Under these conditions the so-called “seeding effect” occurs: the lighter carrier gas molecules accelerate and entrain the heavier, dilute species via collisions, such that all species—regardless of mass—emerge from the expansion with approximately the same terminal velocity v_{∞} . However, this mechanism does not apply to the droplets. Due to their vastly greater size and mass compared to N_2 molecules, droplets experience only limited acceleration during the expansion. Based on literature, droplets that survive the atmospheric-pressure region of the electrospray interface and are transported through the transfer capillary into vacuum, typically have diameters in the range 50–200 nm. Assuming spherical droplets with an average radius of $r_{droplet} = 50$ nm, their mass can be estimated as $m_{droplet} = \rho_{sol} \cdot \frac{4}{3}\pi r_{droplet}^3 = 5.54 \times 10^{-19}$ kg, where $\rho_{sol} = 1.0587 \text{ kg m}^{-3}$ is the density of the polymer solution used in our ESD experiments. The corresponding mass ratio between a droplet and a N_2 molecule is thus $\frac{m_{droplet}}{m_{N_2}} = 1.18 \times 10^7$. This seven-order-of-magnitude difference highlights the inertial mismatch between droplets and gas molecules. As a result, an unrealistically large number of collisions would be required to accelerate the droplets to the speed of the N_2 carrier gas. The seeding effect, therefore, does not occur for droplets under these conditions.

A more realistic estimate of the droplets' final velocity considers the drag acceleration they experience during the early stages of the expansion. For this, we use a simplified form of the Epstein drag force, F_d , which describes the force exerted on spherical particles in high Knudsen number (rarefied gas) flow:

$$F_d \sim \pi r_{droplet}^2 n_{N_2} m_{N_2} v_{rel}^2$$

where n_{N_2} is the number density of N_2 molecules and v_{rel} is the relative velocity between the droplets and the surrounding N_2 gas. The relative velocity changes during expansion but, in order to evaluate an upper limit for F_d , we simplify by assuming that $v_{rel} \sim v_{\infty}$, resulting in

$$F_d \sim \pi r^2 n_{N_2} m_{N_2} v_{\infty}^2$$

The corresponding acceleration experienced by the droplets is:

$$a_d = \frac{F_d}{m_{droplet}} \sim \pi r^2 n_{N_2} v_{\infty}^2 \frac{m_{N_2}}{m_{droplet}}$$

where we have taken the value $n_{N_2} \sim \frac{pN_A}{RT} \sim 2.4 \times 10^{22} \text{ m}^{-3}$, corresponding to a pressure of approximately 1 mbar measured in the first vacuum stage of our ESD system.

This acceleration acts only as long as the expanding N₂ gas retains sufficient density to continue transferring momentum to the droplets. A reasonable estimate is that this coupling effectively ends once the nitrogen reaches its terminal velocity, which occurs over a distance $\Delta x \sim 0.5\text{--}1$ mm (the position of the Mach disc can be taken as an upper limit). The associated time interval is thus $\Delta t \sim \Delta x/v_\infty \sim 1.25$ ms (more generally, something between 0.5-2 ms). The velocity gained by the droplets is therefore approximately:

$$v_{droplet} = a_d \Delta t \sim \pi r^2 n_{N_2} v_\infty^2 \frac{m_{N_2}}{m_{droplet}} \Delta t \sim 12.4 \text{ m s}^{-1}$$

This result indicates that the droplets gain only about 1.6% of the gas velocity, confirming that their acceleration during the expansion is severely limited.

Returning to the original question of the impact kinetic energy per polymer upon collision with the surface, this can be estimated as:

$$E_{impact} = \frac{1}{2} m_{polymer} (v_{droplet})^2 = 1.72 \times 10^{-20} \text{ J} = 0.148 \text{ eV} = 3.41 \text{ kcal/mol}$$

where, as an example, we have used the average mass of polymer **2**, M_n of 134 kDa [A. Onwubiko et al., Nat. Commun. 9, 416 (2018)].

This represents an upper limit for the kinetic energy transferred to each polymer, as it assumes that the entire kinetic energy of the droplet is converted into internal degrees of freedom of its constituent molecules (translations, rotations and vibrations), with no losses due to dissipation mechanisms such as droplet deformation, spreading, splashing or interactions with the substrate. Nevertheless, this energy is over one order of magnitude lower than the typical rotational barrier for a C=C bond. Therefore, it is highly unlikely that the c2 unit arises from rotation around the C=C bond induced during deposition.

In order to address this issue, the following new paragraph has been added to Section 4.2 of the Supplementary Information:

We note that rotation around C=C double bonds is energetically highly unfavourable, with substantial energy barriers (e.g., approximately 65 kcal/mol for the cis–trans isomerization of but-2-ene). These barriers are significantly greater than the kinetic energy imparted to the polymer chains during ESD deposition. Therefore, we can confidently exclude the possibility that the observed coupling defects arise during the ESD process; instead, they must originate from processes occurring during polymer synthesis.

The difference in reactivity between alpha- and beta-carbonyl (as demonstrated in C3 and C1) can be expected for the aldol reaction. Have the authors studied the model reactions? I think this would explain the reactivity and selectivity for the particular oxindole monomers, which also helps to explain the coupling defect ratio in the resulting polymers.

We thank the Reviewer for the suggestion to improve our work. However, we are not fully convinced that a model reaction could help, for the following two main reasons:

i.

Model reactions involving the functional moieties of the relevant monomers have already been investigated in our previous work, in which polymers **1** and **2** were first introduced [A. Onwubiko et al. Nat. Commun. 9, 416 (2018)]. As discussed in the corresponding Supplementary Information, “Significant side products were not observed during the isoindigo condensation reaction.”

ii.

Due to the lower solubility of the conjugated polymer compared to the small molecule monomers, the reaction equilibrium in a model reaction would not necessarily reflect that of the actual polymerisation. In fact, the equilibria of most chain polymerisations and polyeliminations are not governed by complete equilibrium among all reactants [e.g. see H.G. Elias, Thermodynamics of polymerisation. In Macromolecules: Volume 1: Chemical Structures and Syntheses, Wiley-VCH, 193-219 (2005)]. Therefore, it is not evident that the outcome of a model reaction would provide insight into the nature and frequency of the coupling defects observed in the polymer, in particular when these are present in relatively small amounts.

The observation of C=O/CH₂ interconversion via hydride transfer is indeed interesting for the current work. I would suggest that the authors look at the model reactions, which will be very helpful in understanding how the chemistry works. And what about the efficiency of this interconversion in relation to the reaction conditions (acidic or basic condition)?

For similar reasons to those outlined above, we are not fully convinced that a model reaction would accurately represent the processes occurring during polymerisation.

n-type fully fused organic mixed ionic-electronic conductors (OMIECs) are typically synthesised in acidic conditions because these conditions stabilise the electron-deficient backbone. Acidic environments protonate or interact with electron-withdrawing groups, enhancing the solubility and stability of the monomer intermediates during synthesis. This is critical for fully fused systems, which often have rigid, planar structures prone to aggregation or poor solubility. Conversely, basic conditions, can deprotonate or destabilise these electron-deficient groups, leading to side reactions, reduced solubility, or disruption of the conjugation. In fact, in our previous work [A. Onwubiko et al., Nat. Commun. 9, 416 (2018)], we conducted model reactions by heating isoindigo with stoichiometric oxindole under polymerisation and basic conditions, which did not result in the formation of any aldol condensation product. As such, though surely interesting in its own right, we believe that investigating further the observed interconversion under basic conditions is outside the scope of this current study.

The details of the polymer synthesis (4 and 5) should be given in the main text or in the SI. The ¹H NMR spectra for polymer 4 appear to be of poor quality. There are too many dirty peaks and the assignment of the peaks is not clear.

We thank the Reviewer for these constructive suggestions.

The detailed synthetic procedures for all polymers, including polymers **4** and **5**, have now been added to the Supplementary Information in a newly created Section 2.

We also agree with the Reviewer that the original ¹H NMR spectra of polymer **4** were of limited quality, making it extremely difficult to assign the peaks reliably and draw meaningful conclusions. In response, we conducted new variable temperature (VT) ¹H NMR experiments in o-C₆D₄Cl₂ over a temperature range of 5 to 100 °C to obtain higher-quality spectra. In addition, to minimise interference from residual water signals—which can overlap with polymer resonances and compromise peak integration—we performed water suppression experiments. These measurements yielded significantly improved ¹H NMR spectra, enabling accurate integration of the signals corresponding to the alkyl and glycol regions. The resulting alkyl:glycol ratio of 0.80:0.20 is in excellent agreement with the STM data for polymer **4**, as reported in Table S2.

Accordingly, we have completely revised (what now has become) Section 5 of the Supplementary Information to reflect these new, quantitative results. We have also removed the earlier ¹H NMR and CV data recorded in different solvents, as their interpretation was not straightforward and only allowed for qualitative conclusions.

Reviewer #2

In this paper, the authors demonstrate the strength of using STM to characterize semiconducting polymers to give quantitative analysis of the synthesized products. Given that techniques such as NMR can only give an analysis of the average, STM offers a significant advantage. For this reason, I believe that this paper is appropriate for publication in Nature Communications. However, I do have one major concern. The aldol reaction, unlike the cross-coupling reactions that the authors have investigated before, is a reversible reaction. Given the use of high fields during electrospray deposition, how confident are the authors that the results seen by STM are from the synthesis and not of additional reactions that took place during deposition? Without confirming this, any analysis of the polymers would seem premature.

We thank the Reviewer for this important and insightful comment. As they correctly point out, the conjugated polymers investigated in this work differ significantly from those in our previous STM studies in one key aspect: the aldol condensation reaction used to synthesise them is reversible, whereas the cross-coupling reactions employed in our earlier work are not. We therefore understand the relevance of the Reviewer's question, namely whether the defects we observe might arise during the polymerisation process itself or instead be artefacts introduced during the electrospray deposition (ESD) procedure used to prepare monolayer samples for STM imaging in ultrahigh vacuum. While such a question could, in principle, be asked of any polymer or molecule investigated by ESD-STM, it becomes particularly pertinent in this case due to the reversible nature of the aldol condensation reaction.

However, we do not believe that the ESD process can account for the types of defects observed in our STM studies. In fact, we can envisage only two potential mechanisms by which ESD might, in principle, alter the polymer structure:

- a) modifications induced by the high electric field during electrospray; and
- b) structural changes due to the impact of polymer chains upon landing on the Au(111) substrate.

Regarding possibility (a), we note that although the applied voltage between the emitter needle and the entrance capillary in the electrospray deposition (ESD) setup is relatively high (typically 1–2 kV), the solvent system—chlorobenzene and methanol in a 5.6:1 ratio—is not an electrolyte and does not support significant ionic conduction. For a sustained electrochemical reaction to occur, there must be a continuous ionic path between the two electrodes. However, in our ESD setup, once the solution exits the emitter, it rapidly transitions from the Taylor cone into a jet of spatially separated, charged droplets. In this regime, the only conceivable ionic path is through the migration of these lowly charged droplets from emitter to capillary—a mechanism that is highly inefficient for sustaining any appreciable Faradaic current. Furthermore, for electrochemical reactions to proceed, a significant voltage drop must occur at the electrode/solution interface in order to provide the thermodynamic driving force required to align the molecular HOMO or LUMO levels with the Fermi level of the electrode. In conventional electrochemical systems, this interfacial potential drop is localised within the nanometre-scale structure formed by the rearrangement of mobile ions near the electrode surface, the electrical double layer (EDL). The presence of a well-defined EDL is therefore essential for enabling interfacial electron transfer.

In our system, however, the solvent mixture lacks free ions and is overwhelmingly composed of chlorobenzene—a solvent with low dielectric constant (~ 5.7), limited dipole mobility and poor ion-solvating capacity. While methanol is a polar protic solvent capable of supporting double-layer formation, its low proportion ($\sim 15\%$) in the mixture renders its contribution minimal. As a result, the solution cannot sustain a strong or structured EDL and the interfacial potential drop is expected to be weak and diffuse. Without a localised field at the electrode interface, molecular orbital alignment relative to the electrode Fermi level is inefficient, and electron transfer processes are therefore thermodynamically and kinetically unfavourable.

For these reasons, we consider it extremely unlikely that any electrochemical transformations, such as those that could give rise to sequence or coupling defects in the conjugated polymers, occur in the solution phase during ESD.

As for possibility (b), the kinetic energy of the polymer molecules upon deposition onto the Au(111) surface is insufficient to cause structural changes such as *cis-trans* isomerisation (as discussed in the response to Reviewer #1's first question). Consequently, it is even less plausible that such impact events could result in more complex rearrangements or sequence defects.

In light of these considerations, we remain confident that the defects observed in our STM data are intrinsic to the polymerisation process itself and not artefacts of the ESD sample preparation.

Here are additional minor comments:

1. A super minor point – on page 8, it is noted that 3 was obtained by replacing the branched hydrophobic alkyl side chain of 2 with a linear hydrophilic 7-unit ethylene glycol (EG) chain. One other change is that the linear C12 chain has been replaced by a C10 chain according to Figure 1.

We thank the Reviewer for pointing this out. This was indeed an omission on our part for which we apologise. We have now corrected the main text to accurately reflect both structural modifications.

2. For the experimental details of the synthesis of the polymers, the authors refer to previously published work. However, it would be helpful to have the details in this paper too to confirm aspects of their findings. Just as one example, on page 11, the characterization of polymer 4 indicates that comonomer C was not incorporated very much. It would be helpful to immediately have in hand what monomer ratios were used in the synthesis.

We thank the Reviewer for these constructive suggestions. The synthetic procedures for all polymers, including polymer 4, have now been added to the Supplementary Information in a newly created Section 2.

3. Page 11: Can there be a table quantifying the defects in polymers 4 and 5, in the same way that there is one for polymers 2 and 3?

We thank the reviewer for this helpful suggestion. The defect quantification for polymer 4 had already been carried out and is available in the Supplementary Information (Table S2). However, the Reviewer is entirely correct that the corresponding analysis for polymer 5 was missing, and we fully agree that it should have been included. We apologise for this oversight and have now added the relevant data in the Supplementary Information as Table S3.

4. Page 11: The NMR shows that there is ~36% of the hydrophilic monomer. I know that there is error in the NMR analysis but how does this compare against the STM results?

In response to this comment by the Reviewer (and also in response to a related comment by Reviewer #1), we conducted new variable temperature (VT) ¹H NMR experiments in *o*-C₆D₄Cl₂ over a temperature range of 5 to 100 °C to obtain higher-quality spectra. In addition, to minimise interference from residual water signals—which can overlap with polymer resonances and compromise peak integration—we performed water suppression experiments. These measurements yielded significantly improved ¹H NMR spectra, enabling accurate integration of the signals corresponding to the alkyl and glycol regions. The resulting alkyl:glycol ratio of 0.80:0.20 is in excellent agreement with the STM data for polymer 4, as reported in Table S2. These new quantitative results are now described in a completely revised version of (what now has become) Section 5 of the Supplementary Information.

5. Section 4 in SI: Based on the trends in solubility, I would have expected the polymer synthesized in aniline to have the greatest hydrophilic monomer content. Can the authors elaborate on this a bit more? The paper would also be greatly strengthened if the STM results for the polymers synthesized in aniline and aniline:toluene could be included.

We agree with the Reviewer's comment that polymer **4**, when synthesised in a hydrophilic analogue of toluene—namely anisole—would be expected to have a higher content of the more hydrophilic monomer. Indeed, our previous ¹H NMR and CV measurements indicated an increase in the estimated C:A comonomer ratio within polymer **4** when moving from synthesis in toluene to synthesis in a 1:1 toluene:anisole mixture. However, the interpretation of these data was not straightforward and only allowed for qualitative conclusions. The underlying assumption—that increasing solvent hydrophilicity directly favours incorporation of the more hydrophilic comonomer—was based on a simplified view that considers only the hydrophilicity or hydrophobicity of the side chains. In reality, the solubility of each comonomer is influenced also by the structure and chemistry of the conjugated core, making this assumption overly reductive. For this reason, and in light of the much clearer and more quantitative results now available from our new variable-temperature ¹H NMR experiments in o-C₆D₄Cl₂, we have decided to remove the earlier ¹H NMR and CV data collected in different solvents from Section 5 of the Supplementary Information.

Regarding the STM characterisation, we agree that a comparative analysis of polymers synthesised in solvents of varying hydrophilicity would be highly interesting, particularly in combination with high-quality NMR data. Such an approach would indeed further demonstrate the unique analytical capabilities of the ESD-STM technique. However, we believe that this direction lies beyond the scope of the present work and would be more appropriately explored in a dedicated follow-up study.

6. Page 12: "This finding demonstrates that this type of coupling defects is not limited to polymerisation reactions, as is seen in the case of polymers 1-4, but can also occur during the synthesis of small molecules, as seen with polymer 5." For the defects occurring during monomer synthesis, it should be possible to see these in the 1H NMR (if it can't be seen in 1D NMR, it should be possible to see it in the 2D NMR). Can the statement be confirmed with further analysis? Given that the aldol reaction is reversible, it would be good to check that the defect formed during monomer synthesis instead of during the polymerization when decoupling followed by coupling could have taken place.

We thank the Reviewer for raising this important point and fully agree that, based on the currently available data, we cannot definitively conclude that the coupling defects in polymer **5** originate during the monomer synthesis. Prompted by the Reviewer's comment, we have revisited our analysis and interpretation more thoroughly. This has led to what we believe is an important, albeit slightly different, insight: a clearer understanding of how defects arising from the aldol condensation reaction impact the final molecular structure in distinct ways, depending on whether the reaction is employed for small-molecule synthesis or for direct polymerisation.

The NMR characterisation of the TBDOPV monomers used to synthesise polymer **5** has already been reported in our previous work [Y. Cao et al., Chem. Mater. 29, 718 (2017); see Supplementary Information], where no evidence of coupling defects was observed (the spectra are reproduced here

below for convenience). Moreover, the same study also includes the single crystal X-ray diffraction (XRD) structure of TBDOPV, which clearly confirms its *trans* geometry. This is consistent with expectations, as the TBDOPV monomers were purified by silica gel chromatography with eluent (PE:CHCl₃ = 1:8) prior to NMR and XRD analysis and further purified by HPLC before being used in the Stille polymerisation to produce polymer **5**. These preparative chromatography methods are expected to effectively separate the *trans* and *cis* isomers in the small molecules.

Based on our statistical analysis of STM images (now summarised in the newly added Table S3 of the Supplementary Information), polymer **5** exhibits only 1.3% total coupling defects, a level below the detection limit of NMR. Consequently, we cannot conclusively determine whether this very small fraction of defects originates from trace impurities in the monomer (undetectable by NMR), or whether they are introduced during the successive Stille polymerisation step, potentially due to the reaction conditions such as temperature or the presence of acidic/basic species.

However, the key insight prompted by the Reviewer's observation is that the frequency of defects is substantially lower when aldol condensation is used to synthesise small-molecule comonomers (as in polymer **5**) compared to when it is used directly for polymerisation (as in polymers **1-4**). We attribute this difference to the effectiveness of chromatographic purification for small molecules, where *trans* and *cis* isomers differ significantly in polarity and can thus be efficiently separated. In contrast, defective polymer chains—within the observed levels of defective couplings—are not sufficiently distinct from their defect-free counterparts to enable separation by standard purification techniques.

Thus, while defects likely occur during both small molecule and polymer-forming aldol condensation reactions, only in the former can they be efficiently removed. In the latter, they remain embedded in the final polymeric product. These defects could not be previously detected, even when present at levels around 5%, as conventional analytical methods cannot resolve the precise sequence and composition of conjugated polymers. Our study, using ESD-STM, makes such analysis possible.

We have now highlighted this revised and more accurate interpretation in the final part of the results section.

¹H and ¹³C NMR spectra of TBDOPV-318. The asterisk indicates the CH₂Cl₂ peak.

^1H NMR and ^{13}C NMR spectra of TBDOPV (monomer).

Response to reviewers' comments

(2nd round)

We thank the Reviewers for their responses to our rebuttal and for taking the time to re-evaluate our manuscript. We appreciate that both Reviewers acknowledge that we have addressed their concerns—Reviewer #2 in full, and Reviewer #1 for the most part.

The only remaining comment by Reviewer #1 is as follows:

The authors have mostly addressed my concerns in the revised manuscript. However, it is a pity that they did not follow the suggestion for model compound studies. I understand that this would require considerable effort. In light of this, I wonder if the authors could provide computational studies of the reactions or intermediates, which could help to clarify the reactivity and selectivity related to C2 and C3 coupling defects.

We agree with the Reviewer that carrying out a study of model compounds would require considerable effort. However, our main concern is not the complexity of such a study, but rather that it is unclear whether model reactions would offer reliable insight into the coupling defect ratios observed in the corresponding polymers. An investigation based on model reactions would likely yield two main outcomes: (1) confirmation that reactions leading to c2 and c3 couplings (see Figs. 3a, S10, S11, S15) can indeed occur, and (2) determination of the relative frequencies of these coupling defects, which could in principle be compared to the values extracted from STM analysis of the full polymers.

Regarding point (1), given the intrinsic reversibility of the aldol condensation reaction, there appears to be little room for doubt that reactions of the type leading to c2 and c3 couplings can occur. Consequently, we do not believe that performing model reactions or substituting them with computational studies would yield substantial new insight.

Point (2) could, in principle, offer further understanding but presents a different kind of challenge. As noted in our previous response, the conditions in small-molecule model systems differ fundamentally from those governing the polymerisation of conjugated materials. In particular, the limited solubility of the growing polymer chains can shift the reaction equilibrium in ways that are not captured in model reactions. As a result, the defect ratios obtained from such systems may not reliably reflect those present in the actual polymers, especially when the coupling defects occur at low frequency.

Computational studies would likely face similar, if not greater, limitations. Accurately capturing the interplay of solubility, concentration effects and the kinetic versus thermodynamic landscape of the real synthetic conditions would be extremely challenging; yet these are precisely the kinds of factors that are expected to play a key role in determining the distribution of coupling defects.

In light of these considerations, we believe that computational studies would be unlikely to offer substantial additional clarity regarding the factors influencing c2 and c3 coupling outcomes. To help make this point clearer, we have added a brief paragraph to the manuscript (in blue font for better visibility)

noting that model reactions for polymer **2** have previously been reported, but did not yield conclusive results:

Model reactions for polymer **2** have been reported without significant side products;³⁸ while this may reflect limitations in detecting low-abundance byproducts, the much lower solubility of the polymer compared to its monomeric precursors means that reaction equilibria in model systems are unlikely to reflect those of the actual polymerization. In fact, the equilibria of most chain polymerisations and polyeliminations are not governed by complete equilibrium among all components.³⁹ As a result, model reactions are not expected to reliably represent the frequency of coupling defects in the polymer. Only direct structural analysis of the polymers themselves, as performed here, can provide this level of insight.